# Antiaging Potential of Peptides from Underused Marine Bioresources

**DOI:** 10.3390/md19090513

**Published:** 2021-09-10

**Authors:** Enqin Xia, Xuan Zhu, Xuebin Gao, Jindong Ni, Honghui Guo

**Affiliations:** Dongguan Key Laboratory of Environmental Medicine, School of Public Health, Guangdong Medical University, Dongguan 523808, China; xiaenqin@gdmu.edu.cn (E.X.); zxuu1109@gdmu.edu.cn (X.Z.); Gaoxueb@gdmu.edu.cn (X.G.)

**Keywords:** antiaging, underused marine organism, peptides, mechanism

## Abstract

Aging is a biological process that occurs under normal conditions and in several chronic degenerative diseases. Bioactive natural peptides have been shown to improve the effects of aging in cell and animal models and in clinical trials. However, few reports delve into the enormous diversity of peptides from marine organisms. This review provides recent information on the antiaging potential of bioactive peptides from underused marine resources, including examples that scavenge free radicals in vitro, inhibit cell apoptosis, prolong the lifespan of fruit flies and *Caenorhabditis elegans*, suppress aging in mice, and exert protective roles in aging humans. The underlying molecular mechanisms involved, such as upregulation of oxidase activity, inhibition of cell apoptosis and MMP-1 expression, restoring mitochondrial function, and regulating intestinal homeostasis, are also summarized. This work will help highlight the antiaging potential of peptides from underused marine organisms which could be used as antiaging foods and cosmetic ingredients in the near future.

## 1. Introduction

In recent history, human life expectancy has continuously risen, and the proportion of the elderly population has sharply increased with the improvement of medical technology. Dysfunction of normal biological processes accompanies aging, and the acceleration of molecular damage and dysfunction in cells, tissues, and organs eventually leads to aging-related diseases, such as cardiovascular diseases, nutritional metabolic diseases, and cognitive disorders [1,2]. However, the aging process is adjustable, albeit irreversible, and scientific approaches can regulate the speed of aging and promote good health [3]. Therefore, development of efficient strategies to manage aging is critical for community health. In recent years, a growing body of work has aimed to test several hypotheses related to the molecular mechanisms that play critical roles in determining longevity. Among them, reactive oxygen species (ROS) inducing oxidative stress is considered one of the primary contributors to aging [4]. Hence, exploration of substances with antioxidant activity is an important strategy for the management of aging [5,6]. Many kinds of anti-aging drugs have been synthesized and tested in clinical trials, but many obstacles prevent drug development, including the adverse reactions of the human body to synthesized drugs. Therefore, natural active substances with high efficiency and low side effects are important alternatives for combating aging [6].

Recently, natural bioactive peptides have attracted special attention. Biologically active peptides with anticancer, anticoagulant, antidiabetic, antihypertensive, antimicrobial, antioxidant, and cholesterol-lowering properties have been reported in the biomedicine and pharmaceutical biotechnology literature [7]. Some of these compounds also exhibit notable antiaging activity in vitro or in vivo. Presently, potential bioactive peptides are mainly isolated and identified in different food products like milk, soy, rice, meat, fish, lobsters, crabs, and shrimp [8]. Some bioactive peptides are extremely safe, easily absorbed, and have no toxic side effects [9,10]. Moreover, because of their hypo-allergenicity characteristics, bioactive peptides have been explored as a method to treat patients with food allergies [11]. Therefore, bioactive peptides should be considered as excellent alternatives for anti-aging treatment.

Marine organisms contain high-quality functional protein or peptides with diverse molecular structures, and are an important source of new compounds, especially for China with its 18,000 km coastline, compared to terrestrial resources [12]. Presently, excluding various fish that are widely used as food materials, there is only limited use of marine protein resources from other species, such as sea cucumber, sea urchin, mussels, and several kinds of algae. Some marine bioactive peptides regulate free radical homeostasis in vitro and in vivo, and can have antiaging effects in cell and animal models and in human clinical trials [13,14,15,16]. Marine natural bioactive peptides have also been used in cosmeceutical skin products as antiaging agents [17,18,19].

In present work, we used the keywords “peptide” and “anti-aging” or “anti-aging”, we searched the relevant literature published from 1956 to 10 July 2021 from several databases including PubMed, Cochrane Library, Web of Science and Embase. The resultant studies were screened for relevance to underused marine bioresources. Then, we comprehensively reviewed recent studies on antiaging activities of peptides from marine organisms, including their beneficial effects on the regulation of oxidative stress in vitro or in cells, fruit flies, nematodes, mice, and humans. In addition, we summarize the molecular mechanism underlying the antiaging activities of marine peptides. Overall, we aim to highlight useful information for furthering the use of marine sources for bioactive compounds.

## 2. Antiaging Activity of Peptides from the Ocean

Based on the limited literature, marine antiaging peptides have been shown to act in several different ways, including scavenging free radical capacities in vitro, inhibiting cell apoptosis, prolonging the life span of fruit flies and nematodes, and ameliorating D-galactose levels that induce aging in mice, and have shown promise as antiaging agents in human clinical trials.

### 2.1. Free Radical Scavenging Activity and Sequence Characteristics of Peptides

ROS are highly reactive agents in vivo and ROS homeostasis is a key factor in aging. Scavenging free radicals in vitro is a common underlying mechanism among antiaging agents. Therefore, we looked into peptides with free radical scavenging activity in vitro from underused marine resources and reported their 50% effective concentration (EC_50_) values to provide support for their antiaging potential.

#### Free Radical Scavenging Activity

ROS are highly reactive substances, and overproduction of ROS can result in DNA mutation, lipid, and protein dysfunction, and, ultimately, aging [20]. Thus, the main protective strategy to slow aging is to scavenge the excessive ROS by providing antioxidant agents, as well restoring the antioxidant defense system in cells. In the human body, the antioxidant defense system consists mainly of antioxidant peptides, including carnosine, glutathione, and antioxidant enzymes, including superoxide dismutase (SOD), glutathione oxidase (GHS-Px) and catalase (CAT). Therefore, free radical scavenging activity in vitro can reveal potential antiaging activity. Many studies have reported free radical scavenging activity in vitro of marine peptides, and the data are provided in Table 1.

As shown in Table 1, we compiled a list of peptides with antioxidant capacity from several kinds of underused marine organisms, including microalgae and bacteria, several invertebrates, and byproducts of the fishing industry including the head, bone, skin, cartilage, viscera, and gelatin of various fish. In vitro assays show that these peptides scavenge several kinds of free radicals, including superoxide anion (O_2_^−^), peroxide, hydroxyl radical (OH) and 2,2-diphenyl-1-picrylhydrazyl (DPPH) radical, and can affect lipid peroxidation. Evidence for ferric reducing antioxidant power (FRAP, reducing Fe^3+^ to Fe^2+^) is lacking. Short chains with fewer than eight amino acids have been documented as effective antioxidant peptides. According to previous reports, smaller molecular weight peptides are suitable for more easily accessing the reaction center, and preferable for reacting with free radicals to stop the oxidation reaction chain [21]. Peptides with smaller molecular weight also easily penetrate the gastrointestinal barrier to exert their biological effects. In addition, the antioxidant capacity of some peptides from marine organisms can even reach or exceed that of vitamin E and quercetin. For example, the 50% effective concentration (EC_50_) of α-Tocopherol is 0.52 ± 0.03 mM (0.246 ± 0.014 mg/mL) [22]. In comparison, all peptides from microorganisms, invertebrates and tissues of various fish have high antioxidant potential with EC_50_ values of about 1 mg/mL, with no differences between the values in the muscle, skin, and head of fish. These values are about 5–10-fold lower than the EC_50_ value of the cartilage of the red stingray (*Dasyatis akajei*) fish, which performs superoxide anion radical scavenging with an EC_50_ in the range of 0.08–0.16 mg/mL [23]. In addition, for black pomfret (*Parastromateus niger*), peptides from the viscera exhibit stronger free radical scavenging activity than those from the whole body [24]. These results indicate that use of seafood industry by-products and other underused species may be an untapped source of antiaging peptides.

**Table 1 marinedrugs-19-00513-t001:** Marine antiaging peptides and their free radical scavenging activity in vitro.

Source	Sequences	Activities (EC_50_)	Ref.
Microorganism			
*Chlorella vulgaris*	favourzyme hydrolysates	Superoxide^3^ (0.323 mg/mL), Hydroxyl^2^ (0.139 mg/mL)	[25]
*Spirulina* sp.	Thr-Met-Glu-Pro-Gly-Lys-Pro	Inhibition of ROS production	[26]
*Dunaliella salina*	Ile-Leu-Thr-Lys-Ala-Ala-Ile-Glu-Gly-Lys Ile-Ile-Tyr-Phe-Gln-Gly-Lys Asn-Asp-Pro-Ser-Thr-Val-Lys Thr-Val-Arg-Pro-Pro-Gln-Arg	DPPH^1^	[27]
*Penicillium brevicompactum*	N-cinnamoyl tripeptide 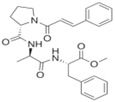	Hydroxyl^2^ (equivalent to that of quercetin at 0.1 mM)	[28]
*Kocuria marina*	Phe-Glu, Asp-Ile, Ser-Ser-Gln, Leu-Glu	DPPH^1^ (0.24 mg/mL)	[29]
Marine invertebrates			
*Neptunea arthritica cumingii*	Tyr-Ser-Gln-Leu-Glu-Asn-Glu-Phe-Asp-Arg	DPPH^1^ (0.77 mM)	[30]
	Tyr-Ile-Ala-Glu-Asp-Ala-Glu-Arg	DPPH^1^ (1.04 mM)	
*Tergillarca granosa*	Glu-Met-Gly-Pro-Ala	DPPH^1^ (0.53 ± 0.02 mg/mL), Hydroxyl^2^ (0.47 ± 0.03 mg/mL), Superoxide^3^ (0.75 ± 0.04 mg/mL), ABTS^4^(0.96 ± 0.08 mg/mL), Inhibition of lipid peroxidation	[31]
	Trp-Pro-Pro-Asp	DPPH^1^ (0.36 ± 0.02 mg/mL), Hydroxyl^1^ (0.38 ± 0.04 mg/mL), Superoxide^3^ (0.46 ± 0.05 mg/mL), ABTS^4^ (0.54 ± 0.03 mg/mL), Inhibition of lipid peroxidation	
*Brachionus rotundiformis*	Leu-Leu-Gly-Pro-Gly-Leu-Thr-Asn-His-Ala,	DPPH^1^ (189.8 µM)	[32]
	Asp-Leu-Gly-Leu-Gly-Leu-Pro-Gly-Ala-His	DPPH^1^ (167.7 µM)	
Fish			
Muscle of *Scomberomorous niphonius*	Pro-Glu-Leu-Asp-Trp	DPPH^1^ (1.53 mg/mL), Hydroxyl^2^ (1.12 mg/mL), Superoxide^2^ (0.85 mg/mL), Inhibition of lipid peroxidation, Protection of plasmid DNA	[33]
	Trp-Pro-Asp-His-Trp	DPPH^1^ (0.70 mg/mL). Hydroxyl^2^ (0.38 mg/mL) Superoxide^3^ (0.49 mg/mL). Inhibition of lipid peroxidation, Protect plasmid DNA.	
	Phe-Gly-Tyr-Asp-Trp-Trp	DPPH^1^ (0.53 mg/mL), Hydroxyl^2^ (0.26 mg/mL), Superoxide^3^ (0.34 mg/mL). Inhibition of lipid peroxidation,	
	Tyr-Leu-His-Phe-Trp	DPPH^1^ (0.97 mg/mL), Hydroxyl^2^ (0.67 mg/mL), Superoxide^3^ (1.37 mg/mL), Inhibit lipid peroxidation.	
Skin of *Scomberomorous niphonius*	Pro-Phe-Gly-Pro-Asp	DPPH^1^ (0.80 mg/mL), Hydroxyl^2^ (0.81 mg/mL), Superoxide^3^ (0.91 mg/mL, ABTS^4^ (0.86 mg/mL), FRAP and Inhibition of lipid peroxidation	[34]
	Pro-Tyr-Gly-Ala-Lys-Gly	DPPH^1^ (3.02 mg/mL), Hydroxyl^2^ (0.66 mg/mL), Superoxide^3^ (0.80 mg/mL), ABTS^4^ (1.07 mg/mL), FRAP and inhibit lipid peroxidation	
	Tyr-Gly-Pro-Met	DPPH^1^ (0.72 mg/mL), Hydroxyl^2^ (0.88 mg/mL), Superoxide^3^ (0.73 mg/mL), ABTS^4^ (0.82 mg/mL), FRAP and inhibit lipid peroxidation	
Cartilage of *Dasyatis akajei*	Ile-Glu-Glu-Glu-Gln	DPPH^1^ (4.61 mg/mL), Hydroxyl^2^ (0.77 mg/mL), Superoxide^3^ (0.08 mg/mL), ABTS^4^ (0.15 mg/mL).	[23]
	Ile-Glu-Pro-His	DPPH^1^ (1.90 mg/mL,), Hydroxyl^2^ (0.46 mg/mL), Superoxide^3^ (0.17 mg/mL), ABTS^4^ (0.11 mg/mL), Lipid peroxidation inhibition activity.	
	Leu-Glu-Glu-Glu-Glu	DPPH^1^ (3.69 mg/mL), Hydroxyl^2^ (0.70 mg/mL), Superoxide^3^ (0.15 mg/mL), ABTS^4^ (0.19 mg/mL), Fe^2+^-chelating ability.	
	Val-Pro-Arg	DPPH^1^ (4.01 mg/mL), Hydroxyl^2^ (1.30 mg/mL), Superoxide^3^ (0.16 mg/mL), ABTS^4^ (0.18 mg/mL).	
Head of *Katsuwonus pelamis*	Trp-Met-Gly-Pro-Tyr	DPPH^1^ (0.33 mg/mL), Hydroxyl^2^ (0.43 mg/mL), Superoxide^3^ (0.38 mg/mL), FRAP and lipid peroxidation inhibition.	[35]
	Trp-Met-Phe-Asp-Trp	DPPH^1^ (0.31 mg/mL), Hydroxyl^2^ (0.30 mg/mL), Superoxide^3^ (0.56 mg/mL), FRAP and lipid peroxidation inhibition.	
	Glu-Met-Gly-Pro-Ala	DPPH^1^ (0.46 mg/mL), Hydroxyl^2^ (0.52 mg/mL), Superoxide^3^ (0.71 mg/mL), FRAP and lipid peroxidation inhibition.	
Salmon gelatin	Gly-Gly-Pro-Ala-Gly-Pro-Ala-Val, Gly-Pro-Val-Ala, Pro-Pro and Gly-Phe	Oxygen radical absorbance capacity (ORAC, 540.94 ± 9.57 µmol TE/g d.w.)	[36]
Pacific cod skin gelatin	Leu-Leu-Met-Leu-Asp-Asn-Asp-Leu-Pro-Pro	Scavenging the intracellular ROS	[37]
Jumbo squid (*Dosidicus gigas*, squid) skin gelatin	Phe-Asp-Ser-Gly-Pro-Ala-Gly-Val-Leu Asn-Gly-Pro-Leu-Gin-Ala-Gly-Gln-Pro-Gly-Glu-Arg	Inhibition of oxidant stress; Lipid peroxidation inhibition (>Vit. E).	[38]
Whole body of *Parastromateus niger*	Ala-Met-Thr-Gly-Leu-Glu-Ala	DPPH^1^ (54%), Metal chelating (78.6%) at 1 mg/mL	[24]
Smooth hound viscera (sharks)	Protein hydrate containing Gly, Glx, Lys, Asx, Arg, Pro and Ala	DPPH^1^, Inhibition of linoleic acid oxidation, Hydroxyl^2^.	[39]
Hoki (*Johnius belengerii*) frame	Glu-Ser-Thr-Val-Pro-Glu-Arg-Thr-His-Pro-Ala-Cys-Pro-Asp-Phe-Asn	DPPH^1^ (41.37 µM), Hydroxyl^2^ (17.77 µM), Peroxyl radical scavenging (18.99 µM), Superoxide^3^ (172.10 µM).	[40]
Limanda aspera frame	Arg-Pro-Asp-Phe-Asp-Leu-Glu-Pro-Pro-Tyr	Inhibition of linoleic acid autoxidation	[41]
Tuna backbone	Val-lys-Ala-Gly-Phe-Ala-Trp-Thr-Ala-Asn-Gln-Gln-Leu-Ser	Inhibited lipid peroxidation, Quenched free radicals (DPPH, hydroxyl and superoxide)	[42]
Frame *of Theragra chalcogramma*	Leu-Pro-His-Ser-Gly-Tyr	Hydroxyl^2^ (35% at 53.6 µM)	[43]

Note: DPPH^1^-DPPH radical scavenging capacity, Hydroxyl^2^-Hydroxyl radical scavenging capacity, Superoxide^3^-Superoxide anion radical scavenging capacity, ABTS^4^-ABTS cation radical scavenging capacity.

### 2.2. Inhibition of Cell Apoptosis

In cell assays, peptides from marine organisms can protect cells injured by ROS induced by several external stimuli, including H_2_O_2_, ethanol, UV-irradiation, and wounding. Cells used in these assays include human umbilical vein endothelial cells (HUVECs), human hepatocellular carcinomas (HepG2), human skin fibroblasts, and human immortalized keratinocytes (HaCaT).

H_2_O_2_ is usually used as an oxidative stress inducer due to its presence in cells and role in redox metabolism. When HUVECs suffer from H_2_O_2_, excessive ROS are generated and deplete intracellular antioxidant enzymes, resulting in apoptosis [44,45,46]. Recently, Oh and coworkers (2021) investigated two cytoprotective peptides, His-Gly-Ser-His and Lys-Gly-Pro-Ser-Trp, derived from seahorse (*Hippocampus abdominalis*). At 600 μM of H_2_O_2_ treatment, high-level intracellular ROS in HUVECs can be detected [47] and apoptotic and necrotic cells are observed [48]. In addition, the cell survival rate of the experimental group decreased by 65.43%. After pretreatment with 100 μg/mL of these two peptides alone or in combination, HUVECs viability was restored to 81.02%, 78.55% and 80.05% for His-Gly-Ser-His, Lys-Gly-Pro-Ser-Trp, and a 1:1 combination of the two, respectively. Surprisingly, the two peptides were not found to exhibit DPPH radical scavenging activity in vitro, suggesting that there are other avenues by which the two peptides counteract oxidative stress induced by H_2_O_2_ in cells [47].

Ethanol is another ROS inducer that acts by increasing cytochrome P450 enzyme 1 (CYP2E1) levels, resulting in oxidative stress and hepatotoxicity in alcohol consumers. The effect of marine peptides from the hydrolysate of the microalgae *Navicula incerta* protein on ethanol-injured HepG2/CYP2E1 cells has been investigated. Cells were pretreated for 1 h with two peptides, Pro-Gly-Trp-Asn-Gln-Trp-Phe-Leu and Val-Glu-Val-Leu-Pro-Pro-Ala-Glu-Leu, at a series of concentrations (0–100 µM) and then treated with 1.0 M ethanol for 48 h. No toxic effect was observed in cells treated with the two peptides at a concentration of 100 mM. The strongest rescue of cell viability by pretreatment with the two peptides was almost 30%. Significant inhibition of gamma-glutamyltransferase (GGT) activity, an indicator of cytotoxicity, was observed. Similarly, both peptides caused concentration-dependent increases in GSH activity. Across all metrics, Pro-Gly-Trp-Asn-Gln-Trp-Phe-Leu outperformed Val-Glu-Val-Leu-Pro-Pro-Ala-Glu-Leu [49]. Thus, these two marine bioactive peptides successfully suppress oxidative stress caused by ethanol.

Ultraviolet (UV) irradiation is a very efficient ROS inducer, especially in human skin, where ROS levels can rise dramatically after only 15-min of UV exposure [50]. There is evidence that marine peptides from fish collagen can partly counteract UV-damage. Human skin fibroblasts pretreated with 0.125%, 0.25% or 0.5% fish collagen for 24 h were UV-irradiated (10 J/cm^2^) for 24 h, and cell viability was restored by 7.9%, 22.3% and 28.1%, respectively. Moreover, mitochondrial activity was restored by 30.6%, 16.4% and 36.1%, respectively [51]. In another cell assay carried out by Han and coworkers, UV irradiation decreased the survival rate of HaCaT cells, illustrated by a total apoptosis rate of 66.8 ± 1.27%. However, after treating injured HaCaT cells with Ile-Cys-Arg-Asp and Leu-Cys-Gly-Glu-Cys, two peptides isolated from Tuna roe by enzymatic hydrolysis, the apoptosis rate significantly decreased and the levels of SOD and GSH-Px increased significantly to more than 60 and 45 U/mL, respectively. Levels of MDA decreased by more than 11 nmol/mL compared to the control group [52].

Wound-recovery activities involve migration and proliferation of keratinocytes, which are essential for tissue healing. In a scratch-assay wounding model in HaCaT cells, 1 μg/mL of PEP, an antioxidant peptide from fermented *Trapa Japonica* fruit, significantly promotes wound healing: a 4.4-fold increase in the recovery of the scratched area was observed in PEP-treated cells. These studies also found that marine peptides can boost wound-recovery by increasing migration and proliferation of keratinocytes [53,54].

### 2.3. Prolonging Lifespan in Model Organisms

Several researchers have reported that marine peptides have the potential to prolong lifespan in *Caenorhabditis elegans* and fruit fly. The fruit fly is frequently employed as an aging model because its genus purity is very high and its lifespan is very short [55]. In addition, it behaves analogously to mammalian hepatocytes in terms of lipid metabolism and transport systems [56]. Moreover, its lifespan is closely associated with oxidative stress, and lipid peroxidation of membranes results in injury of tissues and organs, which might play key roles in the aging process [57]. Aside from studying normal aging, researchers can also induce early aging in the fruit fly by administering D-galactose [55,58]. The effect of peptides from marine sources on simulated aging of the fruit fly has been investigated by Lin and coworkers [59]. Two peptide fractions with MW below 3000 Da from *Stichopus variegates (SVH-PF)* were administered at final concentrations of 1, 4 and 8 g/L. The intervention culture medium fed to fruit flies contained peptide fractions and 40 g/L D-galactose. Compared to the normal controls, SVH-PF/SVH-CAH-PF (4 and 8 g/L) increased the maximum lifespan of fruit flies by 12.3%/11.4% and 24.8%/23.8%, respectively. The increases were 18.1%/24.0% and 21.0%/24.3% compared to the negative control group. These data prove that there is a detectable positive effect of marine peptides on lifespan in D-galactose-induced aging in the fruit fly [59]

Caenorhabditis elegans is attractive model organism for antiaging research, and the effects of marine peptides on its lifespan have also been reported. Sonani et al. administered a dietary supplement with purified phycoerythrin (PE) to normal *Caenorhabditis elegans* to assess its anti-aging potential. PE (100 µg/mL) treatment increased the mean life-span of *Caenorhabditis elegans* from 15 ± 0.1 to 19.9 ± 0.3 days. Degeneration of physiological functions was also moderated by PE treatment. In addition, PE treatment can raise tolerance to thermal and oxidative stress from 22.2 ± 2.5 to 41.6 ± 2.5% and 30.1 ± 3.2 to 63.1 ± 6.4%, respectively, for 5-day-old adults. Excitingly, PE treatment can also affect the levels of a peptide linked to muscle paralysis in an Alzheimer’s disease model [60].

According to Yu et al., total enzymatic hydrolysate and both <3 kDa and >3 kDa hydrolysate fractions from *Sepia esculenta* exhibit significant dose-dependent protective effects in *Caenorhabditis elegans* individuals treated by 50 mM of paraquat at concentrations in the range 0.5–4.0 mg/mL. Two peptides, Asp-Val-Glu-Asp-Leu-Glu-Ala-Gly-Leu-Ala-Lys and Glu-Ile-Thr-Ser-Leu-Ala-Pro-Ser-Thr-Met, obtained from *Sepia esculenta* also exhibit notable antioxidant capacity in *Caenorhabditis elegans* by increasing SOD activity and decreasing the levels of ROS and malondialdehyde (MDA) in paraquat-treated nematodes [61]. Similarly, three novel antioxidant peptides, Leu-Ser-ASp-Arg-Leu-GLu-Glu-Thr-Gly-Gly-Ala-Ser-Ser, Lys-Glu-Gly-Cys-Arg-Glu-Glu-Pro-Glu-Thr-Glu-Lys-Gly-His-Arg, and Ile-Val-Thr-Asn-Trp-Asn-Asn-Met-Glu-Lys, from *Meretrix meretrix* can also rescue paraquat-induced damage in nematodes [62]. Moreover, the favourzyme hydrolysates of *Chlorella vulgaris* protein, which have antioxidant activity, can significantly prolong lifespan, decrease ROS levels, and increase the activity of antioxidant enzymes such as CAT and SOD in *Caenorhabditis elegans* [23].

Aside from toxic substances, excessive glucose intake has also been shown to disturb energy homeostasis and result in the accumulation of fat in *Caenorhabditis elegans*, which promotes the generation of ROS and even induces lipid peroxidation and oxidation-damage [63]. After pretreatment with two peptides, Asp-Val-Glu-Asp-Leu-Glu-Ala-Gly-Leu-Ala-Lys and Glu-Ile-Thr-Ser-Leu-Ala-Pro-Ser-Thr-Met, obtained from *Sepia esculenta*, the levels of fat accumulation and lipid peroxidation induced by 10 mM glucose were significantly decreased, while ROS and MDA levels were restored [61]. In addition, Herring milt protein hydrolysates can also improve glucose homeostasis in obese mice fed a high-fat diet [64].

### 2.4. Ameliorating D-Galactose Induced Aging in Mice

In antiaging studies, mouse models of aging can be induced by D-galactose [65]. During induction, D-galactose causes accumulation of ROS and an increase in oxidative stress and damage to cell membrane lipids [66]. The earliest damage is found in the liver. Then, a series of pathogenesis linked to aging and metabolic syndrome are observed in succession. Therefore, metabolic disorder and oxidant stress are considered the two key factors for aging. To make matters worse, disorders of glucose and lipid metabolism can lead to susceptibility of cells and tissues to oxidative stress.

Zhu et al. showed that after treatment with D-galactose (200 mg/kg), mice age and exhibit more missing hair and whiskers, obvious kyphosis, and abdominal obesity, and significantly higher mean senescence scores compared with control mice. Mice treated with mussel peptides have more hair and bushy whiskers, as well as lower body weight and liver indexes, improvements to disorders of glucose and lipid metabolism, increased antioxidant capacity, and slowed aging [67].

### 2.5. Antiaging Effects in Human Clinical Assays

Peptides obtained by hydrolysis, isolation and purification from fish and other food material exhibit antiaging potential. Recently, in a randomized crossover intervention study, a lean-seafood diet was found to impact mitochondrial energy metabolism of healthy human subjects based on detection of urinary markers. In the study, the intervention group ate a diet with several lean fish, including cod, pollack, saithe and scallops, and the control group ate a diet of land protein including chicken, lean beef, turkey, pork, egg, milk and milk products. Twenty healthy human subjects out of 148 subjects completed the 8 weeks test. Results showed that the lean-seafood intervention significantly decreased the levels of carnitine, an acylcarnitine and N1-methyl-2-266 pyridone-5-carboxamide (2PY) in urinary samples, indicating that mitochondrial function was improved [68]. The data suggest that for healthy subjects, a lean seafood diet of cod, pollack, saithe, and scallops might increase antioxidative capacity of mitochondria, boost normal lipid catabolism, and lower oxidative stress [68]. Clinical studies examining diets containing potential anti-aging peptides from marine fish bioresources and peptides from underused species are an important avenue for further study.

Apart from oral administration assays, studies on skin aging using cosmetics with marine peptides have also been carried out [69]. In terms of underused marine sources, PEP, a synthesized peptide based on the structure of the antioxidant polypeptide from *Trapa japonica* fruit extract, has been recently evaluated at the clinical trial level for antiaging effects on skin. The intervention employs an eye cream at 0.5% concentration. Results showed that 0.5% PEP intervention for eight weeks resulted in significantly less skin wrinkling [64].

Collagen synthesis is considered a key factor in skin aging [53,70]. Sufficient amino acids and oligopeptides binding to the surface of fibroblasts are both important for collagen synthesis [71]. Therefore, oral supplementation of low molecular weight hydrolyzed collagen (0.3–8 kDa) can prevent aging by providing ample material for collagen synthesis by human skin fibroblasts. The results indicate that small molecular weight peptides from collagen support collagen synthesis and improve the structure of extracellular matrix (ECM). In line with this, several studies have reported that small peptides from hydrolyzed collagen can stimulate the formation of ECM proteins, boost fibroblast proliferation, and slow aging after UV-irradiation and other harmful external environmental stimuli [14,16,72].

## 3. The Mechanisms Underlying Antiaging Activity

Marine peptides that exhibit free radical scavenging activity in vitro (Table 1) have potential antiaging effects. However, other mechanisms may also be involved in slowing aging in vivo, including an increase in the activity of antioxidant enzymes, regulation of Klotho, inhibition of apoptosis, inhibition of matrix metalloproteins-1 (MMPs-1) expression, protection of mitochondrial activity by correct protein folding and decreased DNA mutation by the chaperonin containing t-complex polypeptide 1 and the phosphatase and tensin homolog (CCT-PTEN) Pathway, restoration of intestinal homeostasis and regulation of aging-related metabolic disorders.

### 3.1. Improvement of Antioxidant Enzyme Activity

Antioxidant enzymes are an important target of peptides in cells and organisms. The underlying mechanisms involved include regulation of the classical antioxidant gene Klotho and the Keap1/nuclear fac-tor erythroid 2-related factor 2- antioxidant responsive element (Nrf2-ARE) signaling pathway. In addition, marine peptides can regulate DAF-16/FOXO-SOD-3 expression, i.e., a forkhead transcription factor/forkhead transcription factors of the O class, which can enhance antioxidant enzyme activity.

#### 3.1.1. Regulation of Keap1/Nrf2-ARE Expression

Nrf2, a multidomain repressor protein inhibited by binding to Keap 1, is an essential transcription factor that regulates the expression of an array of detoxifying and antioxidant defense genes. Nrf2 and Keap1 are co-expressed in many organs. Upon stimulation by ROS, for example after UV irradiation or exposure to H_2_O_2_, Nrf2 is released from Keap1 and activated. Free Nrf2 can enter the cell nucleus and combine with antioxidant response elements (AREs). Then, the complex boosts the transcriptional activity of antioxidant enzymes. Simultaneously, the transcription levels of Keap1 and Nrf2 are increased. The Keap1/Nrf2-ARE pathway, as an important antioxidant pathway, plays a significant role in regulating the levels and activities of antioxidant enzymes such as GSH-Px and SOD [73]. Natural peptides from marine sources have been found to protect cells from damage by targeting this pathway.

Two peptides obtained from Tuna roe, Ile-Cys-Arg-Asp and Leu-Cys-Gly-Glu-Cys, have strong DPPH radical scavenging activity in vitro and lead to significant rescue of HaCaT cell exposed to ultraviolet B (UVB) radiation [52]. Further, transcription of antioxidation-related genes, such as Keap1, Maf, CAT, glutathione S-transferases (GST), and Mn-SOD are restored to normal levels, while the transcription of Nrf2, Cu-SOD, and GSH-Px increases to significantly higher levels than those in the model group, with stronger effects observed for the Leu-Cys-Gly-Glu-Cys peptide. Thus, these two peptides protect HaCaT cells irradiated by UV via this classical redox pathway [52].

The antiaging properties of the Leu-Cys-Gly-Glu-Cys peptide have been verified in healthy mouse models. After treatment with the peptides, mice display a healthier status based on a series of body indices, such as increased body weight and decreased liver and brain indices. Based on the analysis of liver and serum samples, the peptide was found to increase the activity of serum GSH-Px and SOD and decrease levels of MDA. Additionally, the transcription levels of Keap1 and Nrf2 were significantly downregulated in the liver and brain. These results confirmed that marine bioactive peptides can be used as potential regulators of aging in cells and mice, extending the earlier findings that showed it can scavenge free radicals in vitro. These results also highlight the importance of the Keap1/Nrf2-ARE pathway in antiaging [52].

Using immunofluorescence assays, Cai and coworkers showed that after treating H_2_O_2_-injured HUVECs with the Phe-Pro-Tyr-Leu-Arg-His peptide from the swim bladder of the miiuy croaker (*Miichthys miiuy*), Nrf2 translocation and accumulation in the nucleus are both down-regulated. In addition, expression of heme oxygenase isozyme-1 (HO-1) increases significantly. These data indicate that Nrf2 nuclear translocation induces HO-1 overexpression, thus preventing apoptosis of H_2_O_2_-injuried HUVECs [74].

#### 3.1.2. Regulation of Klotho

Klotho, known as an aging gene in mammals, regulates several molecular processes including phosphate homeostasis, insulin signaling, and Wnt signaling [75,76]. Moreover, tumour suppressor p53/p21, cyclic adenosine monophosphate (cAMP), and protein kinase C (PKC) are influenced by Klotho. Klotho deficiency leads to susceptibility to several aging-relative diseases and vasculogenesis [77,78,79]. Thus, regulation of Klotho expression is an important strategy for antiaging studies.

A *Stichopus variegates* peptide fraction, SVH-PF/SVH-CAH-PF, contains a considerable number of small peptides with MW below 3000 Da [59]. A series of changes are observed in the liver, brain, and serum of mice exposed to D-galactose at a daily dose of 100 mg/kg, as well as a decrease in the activities of SOD and GSH-Px and an increase in MDA and protein carbonyl levels. Combining D-galactose treatment with orally administered SVH-PFS and SVH-CAH-PF at doses of 200, 500 or 1000 mg/kg for eight consecutive weeks enhances the activities of SOD and GSH-Px in the liver and brain of aging mice, suggesting that the deterioration induced by D-galactose is prevented. In addition, MDA levels decreased (vs. negative control) by over 50% in the liver and 23% in the brain after treatment with SVH-PFS/SVH-CAH-PF at 1000 mg/kg. At a dose of 1000 mg/kg, SVHPF/SVH-CAH-PF administration inhibited the formation of protein carbonyls in the liver, brain, and serum sufficiently to keep them at normal levels. Furthermore, pretreatment with both SVH-PF and SVH-CAH-PF at doses of 500 and 1000 mg/kg efficiently increased Klotho expression in brain [59]. Simultaneously, the peptide fractions effectively promoted longevity and attenuated oxidative injury in D-galactose-treated fruit flies by boosting Klotho expression [59].

#### 3.1.3. Regulation of DAF-16/FOXO SOD-3 Expression

DAF-16, a forkhead transcription factor, is a *Caenorhabditis elegans* orthologue to the mammalian FOXO proteins which regulates stress resistance and longevity [80,81]. The effects of three *Meretrix meretrix* antioxidant peptides, Leu-Ser-ASp-Arg-Leu-GLu-Glu-Thr-Gly-Gly-Ala-Ser-Ser, Lys-Glu-Gly-Cys-Arg-Glu-Glu-Pro-Glu-Thr-Glu-Lys-Gly-His-Arg and Ile-Val-Thr-Asn-Trp-Asn-Asn-Met-Glu-Lys, on the expression of the relevant genes has been investigated. First, in the transgenic *Caenorhabditis elegans* strain GR1352 in normal conditions, 4.0 mM of peptides can significantly increase DAF-16: green fluorescent protein (GFP) localization to cell nuclei. Next, when the transgenic *Caenorhabditis elegans* strain CF1553 expressing a SOD-3: GFP reporter was treated by 4.0 mM of three peptides, the level of SOD-3 significantly increased. Further investigation showed that the sod-3 transcript level increased more than two-fold, and the ctl-1 and ctl-2 transcript levels also increased about two-fold after treatment with the Leu-Ser-ASp-Arg-Leu-GLu-Glu-Thr-Gly-Gly-Ala-Ser-Ser peptide. *Meretrix meretrix* peptides enable nematodes to resist oxidative stress induced by paraquat via up-regulation of DAF-16 [62]. In addition, two antioxidant peptides, Ala-Ala-Val-Pro-Ser-Gly-Ala-Ser-Thr-Gly-Ile-Tyr-Glu-Ala-Leu-Glu-Leu-Arg and Asn-Pro-Leu-Leu-Glu-Ala-Phe-Gly-Asn-Ala-Lys, from *Strongylocentrotus nudus,* also enhance nuclear translocation of DAF-16 and the expression sod-3 in nematodes damaged by oxidant stress [82].

The mechanisms of action of two antioxidant peptides, Asp-Val-Glu-Asp-Leu-Glu-Ala-Gly-Leu-Ala-Lys and Glu-Ile-Thr-Ser-Leu-Ala-Pro-Ser-Thr-Met from *Sepia esculenta*, that promote survival of *Caenorhabditis elegans* treated with paraquat have also been investigated. Transgenic CF1553 nematodes expressing sod-3p:GFP were used [61]. After the nematodes were treated with the two peptides, expression of the antioxidant enzyme genes, sod-3 and cat-1, were significantly increased from baseline levels. Further, the authors showed that these marine peptides protected oxidation-damaged worms via the p53 signaling pathway and sod-3 and cat-1 [61]. However, dietary supplementation with phycoerythrin (PE, 100 µg/mL) extends lifespan in mutant *Caenorhabditis elegans* with knock-out of upstream (daf-2 and age-1) and downstream (daf-16) regulators of insulin/IGF-1 signaling (IIS). These results show that PE does not act through the DAF-2-AGE-1-DAF-16 signaling pathway [60]. More work is required to uncover the underlying mechanisms in this case.

### 3.2. Inhibition of the Autophagy and Apoptosis

As summarized in Table 1, marine antioxidant peptides have potential as antiaging agents because they can protect cells from oxidative stress by directly scavenging excess free radicals. In addition, marine peptides can inhibit apoptosis via two pathways: the p53-Bax/Bcl-2 (BCL2-associated X protein/B lymphoblastic cells) pathway and the Mammalian target of rapamycin complex 1 (mTORC1)-mTOR-Bcl/Bax pathway.

#### 3.2.1. Directly Scavenging Excess Free Radicals

Mitochondria are an important producer of ROS. Its ROS production pathways include the electron transport chain, xanthine oxidase pathway, phospholipase activated arachidonic acid metabolism pathway, and arginine nitric oxide synthase pathway [83]. Under unhealthy conditions, the levels of ROS lose their dynamic balance, resulting in cell damage, cell aging, cell apoptosis, etc [84]. With strong oxidative activity, excessive ROS can lead to DNA fragmentation, damage to the integrity of the cell membrane, and damage to the structure and function proteins. Several studies have shown that high concentrations of ROS can initiate apoptosis. In addition, carcinogens such as aflatoxin B1, bisphenol A, and heterocyclic amine lead to an increase in intracellular ROS levels throughout the process of tumorigenesis [85]. The peptides listed in Table 1 could therefore inhibit apoptosis by protecting mitochondria. The mechanisms involved are displayed in Figure 1.

#### 3.2.2. Regulation of p53-Bax/Bcl-2 Pathway

Apoptosis involves morphological changes, such as cell shrinkage and chromatin condensation, and is regulated by several proteins including p53, caspase-3, and bcl-2 [48]. Mitochondrial DNA damage caused by oxidative stress can activate p53, which then executes the cell apoptosis program [86]. When p53 is activated by oxidative stress, there is an increase in bax expression and a decrease in bcl-2 expression, which increases the Bax/Bcl-2 ratio. The relative expression of Bax and Bcl-2, a pair of proteins that promote and inhibit apoptosis, governs the mitochondrial release of cytochrome C (Cyt-C) into the cytosol [48]. At the same time, oxidative stress induced by H_2_O_2_ increases the permeability of the mitochondrial membrane. These two factors synergistically enhance the release of Cyt-C into the cytosol. Caspase-9 is activated by elevated Cyt-C concentration in the cytosol. Subsequently, caspase-9 activates caspase-3 by cleavage. Finally, cleaved caspase-3 is activated, and the cell carries out apoptosis [87].

The levels of p53 and caspase-3 and the Bax/Bcl-2 ratio induced by H_2_O_2_ decrease by 0.86-, 2.33- and 1.47-fold, respectively, when H_2_O_2_-treated HUVECs are treated with blue mussel (*Mytilus edulis*) hydrolysate (BMEH) at a concentration of 0.5 mg/mL. BMEH significantly decreases H_2_O_2_-mediated apoptosis of HUVECs by decreasing the expression of p53 and caspase-3 genes and the Bax/Bcl-2 ratio (Figure 1) [45].

#### 3.2.3. Regulation of mTORC1-mTOR-Bcl/Bax Apoptosis Pathway

mTOR is a serine/threonine kinase that is a part of the PI3K-related kinase family, an apoptosis activating group [88,89,90]. As raptor decreases, mTOR is released from the mTORC1 complex. Then, dephosphorylation of mTOR and oxidative stress lead to an increase in the Bcl/Bax ratio [90,91]. According to Nam et al., dihydrotestosterone (1 mg/mL) can significantly decrease the levels of phosphorylated-mTOR and raptor in human dermal papilla cells, leading to increased dephosphorylation of mTOR, initiation of apoptosis, and autophagy of mitochondria and membranes. Treatment with the peptide from *Trapa japonica* fruit (10 mg/mL) inhibits cleavage of the mTORC1 complex and restores cell viability. These data suggest that marine peptides can inhibit cleavage of the mTORC1 complex, allowing them to suppress autophagy and apoptosis in injured human dermal papilla cells [92,93] (Figure 1).

### 3.3. Regulation of the TNF-α-MMPs-ECM Pathway to Suppress MMP-1 Expression

Imbalance of the ECM structure and dysfunction of fibroblasts cause the skin aging. Fibroblasts residing in the dermis just under epidermis build the ECM. ECM is mainly composed of collagens, among which type I collagen accounts for 80% of total collagen. Collagens are important for skin elasticity, flexibility, and tension, and are digested by matrix metalloproteinases (MMPs) [94,95]. MMP-1 (collagenase-1) is responsible for degeneration and inhibition of type I collagen synthesis [96,97]. Tumour necrosis factor-alpha (TNF-α), a major inflammatory cytokine, activates MMP-1 and MMP-9 expression and inhibits collagen synthesis in human dermal fibroblasts (HDFs) [98].

Recently, Jang et al. treated HDFs with 1 μg/mL of PEP, a peptide synthesized based on the sequence of a peptide from fermented *Trapa Japonica* fruit, prior to treatment with 20 ng/mL of TNF-α. PEP restored normal levels of MMP-1 and MMP-9 induced by TNF-α. In addition, collagen synthesis increased by 95% compared to controls [53].

### 3.4. Regulation of CCT-PTEN Pathway to Protect Mitochondria

Mitochondrial activity is essential for proper cell function. A decline of mitochondrial activity is associated with division, migration, and aging of cells, and accompanies the onset of several diseases. Very recently, multiple studies reported that several marine peptides can restore or improve mitochondrial function after damage by exposure to external stimuli such as H_2_O_2_ and UV-radiation [45,47,99,100].

UV-irradiation of human skin fibroblasts results in cell division, migration, and apoptosis due to protein dysfunction by misfolding and mitochondrial DNA lesions. CCT, containing CCT1-CCT8 subunits, forms a chaperone complex with tailless complex polypeptide 1 (TCP1) to ensure correct protein folding [101]. PTEN-induced kinase 1 (PINK1)/Parkin-mediated mitophagy can counteract the effects of mitochondrial DNA mutation in flies [102]. After 24 h UV treatment (10 J/cm^2^), collagen peptides from the hydrolysis of fish collagen protein restore the expression levels of CCT2, CCT5, CCT6A, CCT8, and PINK1 genes to normal levels. The collagen peptides not only enhanced the expression of elastin and collagen and relieved the damage caused by ROS, but also boosted protein folding and DNA repair, lowering the risk of DNA mutation in UV-treated fibroblasts [51].

### 3.5. Restoration of Intestinal Homeostasis and Regulation of Aging-Related Metabolic Disorders

Gut microbiota, including both beneficial and harmful flora, is an important part of the human gastrointestinal tract. Beneficial microbiota can contribute to the health of the intestinal tract by ensuring complete absorption and utilization of the diet nutrients, synthesizing some nutrients and bioactive substances, consuming harmful substances, and controlling population levels of harmful bacteria, which can produce harmful metabolites and damage the health of the host [103,104]. *Drosophila melanogaster* models have shed some light on the animal–microbial symbiosis and can be used to test the impact of active compounds on the gut microbiome [105]. Based on the microbial differences between *Drosophila melanogaster* and other species, it appears that microbial content is independent of species consistency. It is, however, closely associated with diet, and similar diets lead to similar microbial composition even among different species [106,107]. Diet and metabolism, and not microbe species, influence the composition of gut microbiota. In addition, significant changes in intestinal microbiota are observed as severe age-related physiological diseases, especially metabolic diseases, develop, suggesting that age-related intestinal changes may seriously affect the overall health and lifespan of the host [107,108,109]. Studies have unveiled the mechanisms by which gut microbiota enhance antioxidant capacity in vivo [110,111]. For example, *Lactobacillus* has been shown to consume metal ions and prevent metal ion oxidation [112]. The gut microbiome is closely linked with host aging, and therefore, treatments that regulate the composition and metabolism of gut microbiota can improve health and slow aging. Data concerning the impact of antioxidant peptides derived from marine sources on gut microbiota are summarized in Table 2.

As shown in Table 2, recent investigations have focused on the impact of marine peptides on gut microbiota. The test substances have included protein hydrolysates, polypeptide fractions, and pure antioxidant peptides. Significant positive effects have been observed in many assays using peptides from microalgea, invertebrates, and byproducts of the fishing industry, such as the milt, roe and viscera of fish. Even glycosylated fish has been shown to have positive results. The balance of gut microbiota populations significantly improves not only in healthy mice or rats, but also in unhealthy ones subject to fatigue, alcohol-induced injury, and high-fat diet (HFD). In addition, a peptide combined with calcium significantly increases the abundance of beneficial gut microbes, including *Firmicutes* and *Lactobacillus*, in low-calcium diet-fed rats. Calcium absorption also increased in these animals.

In a cross intervention study, 20 healthy subjects ate two diets with different protein sources. Non-seafood protein diet induced a decrease in the relative abundance of *Clostridium cluster* IV and increases in the *Firmicutes*/*Bacteroidetes* ratio and the microcapsule/bacteroidete ratio of *Cunninghamia lanceolata*. Compared to the diet of cod, pollack, saithe, and scallops, the diet of chicken, lean beef, turkey, pork, egg, milk and milk products is preferred by beneficial gut microbes. Further examination revealed that circulating triacylglycerol (TAG), total to high-density lipoprotein (HDL) cholesterol ratio, and circulating trimetlylamine oxide (TMAO) levels are each linked to specific intestinal microbes. The fish diet released high levels of TMAO into the circulatory system, and the composition and activity of gut microbiota is regulated by TMAO rather than by active peptides from fish muscle. This study showed that, in the human body, the role of a single antioxidant peptide on gut microbiota is difficult to disentangle from diet-wide effects, and that many factors must be considered during experiment design [68]. As mentioned above, the free radical scavenging activity of peptides from fish viscera and cartilage exhibit higher activity than those from fish muscle. Peptides from other underused fish tissues may play significant roles in the regulation of gut microbiota for anti-aging purposes. *Drosophila melanogaster* is a useful model for studying the interactions between non-pathogenic microbes and the host because it can be genetically and experimentally manipulated [105]. In addition, it can provide an integrative approach to study the relationships between active compounds and the gut microbiome. However, no reports on the impacts of marine peptides on the microbiome of *Drosophila melanogaster* were found.

## 4. Conclusions and Perspectives

This review provides a comprehensive overview of the antiaging activities of marine peptides from underused resources, including the head, bone, skin, cartilage, viscera and gelatin of various fishes, several kinds of microalgae, mussels, rotifers and *Trapa japonica*. The potential antiaging activities of natural peptides from underused marine organisms have been investigated in vitro, in cells, in animal models and in human clinical trials. These peptides act using the following molecular mechanisms: free radical scavenging, enhancing oxidase activity, protecting mitochondria, downregulating the apoptotic pathway, inhibiting MMP-1 expression, and restoring intestinal homeostasis. These results suggest that underused marine peptides have great potential as functional food and cosmetic ingredients for antiaging purposes. However, presently, there are three major outstanding issues. First, relevant research reports in the Web of Science database are few, and not enough animal assays have reported. We therefore focused on in vitro assays for this review. Second, in the limited literature, very few marine species have been explored, and vast marine protein resources are still underutilized. Third, apart from several applications in skin aging, there is a dearth of information about marine peptides in human clinical trials related to antiaging. This lack of literature makes it difficult to apply and transform these peptides for the market. To address these issues, further exploration of the abundant underused marine peptide sources is required.

## Figures and Tables

**Figure 1 marinedrugs-19-00513-f001:**
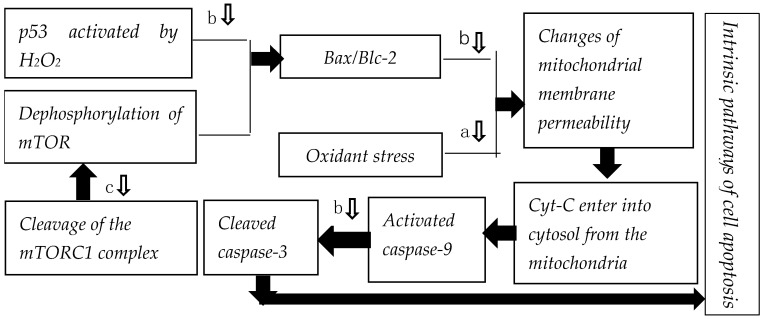
Marine peptides intervene in the intrinsic pathway of cell apoptosis (Note: a—scavenging free radical peptides in Table 1, b—*Mytilus edulis* hydrolysate, and c—peptide from *Trapa japonica* fruit).

**Table 2 marinedrugs-19-00513-t002:** Regulation of gut microbiota by marine peptides.

Marine Peptides	Test Animal	Improvement of Gut Microbiota	Ref.
*Spirulina Phycocianin* (Microalgae)	Mice	Increase the relative abundance of Bacteroidetes and Actinobacteria	[113]
Glycosylated fish protein	Mice	Increase the abundance of *Allobaculum*, *Akkermansia, Lactobacillus animalis*	[114]
Walleye Pollock skin	Mice	Upregulation relative abundance of *Lactobacillus* and *Akkermansia,* downregulation the abundance of bacteria associated with intestinal inflammation	[115]
Skin collagen peptide of *Salmon salar* and *Tilapia nilotica*	Male rats	Increased abundance of *Lactobacillus*	[116]
Herring milt hydrolysate (protein: 47–94%)	Mice	Maintain abundant of *Lactobacillus* decrease metabolites associated with obesity and inflammatory disease	[117]
Peptides from tuna roe	Mice	Short-chain fatty acids production in feces and modulating gut microbiota composition	[52]
Abalone viscera	Alcohol induced injured mice	Increase in diversity index and the number of Bacilli (class), *Lactobacillales* (order), *Lactobacillaceae* (family), and *Lactobacillus* (genus) levels	[118]
*Spirulina platensis* protease hydrolyzate	High-fat diet (HFD)-fed rats	Enriched the abundance of gut beneficial bacteria	[119]
*Chlorella pyrenoidosa* protein hydrolysate-calcium chelate	Low-calcium diet-fed rats	Improving the abundances of *Firmicutes* and *Lactobacillus*	[120]
Oyster polypeptide (OP) fraction	Exhaustive fatigue mice	regulate the abundance of gut microbiota and maintain its balance	[121]

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
