# Peer review of "Antiaging Potential of Peptides from Underused Marine Bioresources"

_marinedrugs, 2021, doi:10.3390/md19090513_

Round 1

Reviewer 1 Report

While the submitted manuscripts tries to provide a comprehensive review for the potential application of marine bio resource-derived peptides for cosmetic and functional foods, unfortunately, it fails to deliver the clear information to readers. Most of all, there are too many inappropriate expressions and typo errors. Also, the editing is also poor and should be revised. Reviewer recommends thorough revision and editing through the whole manuscript. 

Author Response

According to your kind suggestion, the whole manuscript has been thoroughly revised, including content, structure, as well expressions and typo errors, by a senor researchers. please check the revised manuscript. Thank you very much for your help.

Reviewer 2 Report

This study is reviewing the potential anti-aging activity of marine-derived peptides. Although this review paper covers an interesting topic, it has too many problems with its writing. Therefore, it is difficult to publish this review paper in its current form.

  1. The title is too exaggerated. There is no sufficient evidence to support the anti-aging effect of peptides.
  2. There are too many serious mistakes in sentence construction and basic grammar. In order to improve the quality of this review paper, please ask for the help of a senior researcher and a review by a professional English editor.
  3. The contents of Tables 1 and 2 are neither consistent nor systematic. Therefore, it is not clear what information is being conveyed.
  4. There is no experimental basis for Figure 1 and it is not informative.

Author Response

According to your kind suggestion, the whole manuscript has been thoroughly revised, including content, structure, as well expressions and typo errors, by a senor researchers. please check the reply items and the revised manuscript. Thank you very much for your help.

Reviewer 3 Report

The manuscript prepared by Enqin Xia et al. quite broadly summarizes the results obtained in the analysis of the antioxidant and anti-apoptotic properties of peptides from underused marine bioresources, as well as in an interesting statement sums it up. However, it requires some improvements:

  1. the whole structure is very disordered , e.g. when authors start point 2 “Antiaging Activity of Peptides”with their free radical scavenging capacities (2.1) also the point 3 “The Mechanism of Marine Peptides Antiaging Activities” should start wit the mechanism of free radical scavenging (it should be 2.1 and now is 4). Next the antioxidant enzymes activity and apoptosis should be described in a respective way.
  2. the mechanism of marine peptides free radical scavenging capacities should be described in details.
  3. in the case of apoptosis, a diagram/scheme showing which pathways are inhibited by which peptides would be useful.
  4. the abstract is non-informative regarding the description of the peptides that are included in the review.
  5. peptide sequences are abbreviated using either 3 or 1 letter abbreviations maybe it can be standardized?
  6. Line 526 – why (TMAO) is in the brackets?

Author Response

(The authors gave the same response as above.)

Author Response

The reply to the comments and the second revision version have been uploaded. Please check them.

Reviewer 5 Report

The review is very interesting and well written manuscript. However, some commets should be addresed before their publication.

L13. It seems more appropriate to use "clinical" instead "clinic"

L41,56. It seems more appropriate to use "natural" instead "nature"

L46. In order to support the affirmation. Please add the follow recent references: 

  • Cosmetics 20185(1), 21; https://doi.org/10.3390/cosmetics5010021
  • Peptides 2019, 122, 170170; https://doi.org/10.1016/j.peptides.2019.170170

L70,74,91. Please define the acronyms ROS, SOD, GHS-Px, CAT, EC50. Please check through the entire manuscript 

L55, 56, 59, 66, 75, 80, 95,97. Please use italics in latin word and scientific names. Please check through the entire manuscript

Please before the section 6 add an aditional section about Safetyof antiaging peptides, considering their alergenicity and toxicity. I mean, it has been reported that peptides have low allergenicity and are nontoxic for their use in cosmetic and cosmeceutical (Peptides 2019, 122, 170170; https://doi.org/10.1016/j.peptides.2019.170170).

Please re-organize the order of the references. The reference 1 is repeat

Author Response

According to your kind suggestion, the whole manuscript has been thoroughly revised, including content, structure, as well expressions and typo errors, by a senor researchers. please check the reply items and revised manuscript. Thank you very much for your help.

Round 2

Reviewer 1 Report

While the revised manuscript improved significantly, there are still too many points of revision requiring thorough editing. Many sentences are inappropriate and fail to deliver clear meaning. Reviewer strongly suggests a professional, native English editing service for being accepted for publication. 

In Figure 1, reviewer could not understand the exact meaning of the Figure. What is anti-oxidant peptides from terrestrial and marine sources? Is this figure comparing the general property of marine peptides and terrestrial peptides? 

Author Response

The native English edition have been finished. The Figure 1 has been cancelled. please check the second revision version.

Reviewer 2 Report

I could not find a reason to change my first decision.

Author Response

The native English edition have been finished. And the Figure 1 has been cancelled. please check the second revision version.

Reviewer 3 Report

After the changes, the manuscript looks much better, however, its structure (the order of chapters) is questionable and gives the impression of being unwanted.

Author Response

The Figure 1 has been cancelled. please check the second revision version.